# Reading between the (Genetic) Lines: How Epigenetics is Unlocking Novel Therapies for Type 1 Diabetes

**DOI:** 10.3390/cells9112403

**Published:** 2020-11-03

**Authors:** Ammira-Sarah AL-Shabeeb Akil, Laila F. Jerman, Esraa Yassin, Sujitha S. Padmajeya, Alya Al-Kurbi, Khalid A. Fakhro

**Affiliations:** 1Department of Human Genetics, Sidra Medicine, Doha P.O. Box 26999, Qatar; Lali@sidra.org (L.F.J.); esraayassin@gmail.com (E.Y.); ssubashpad@sidra.org (S.S.P.); aalkurbi2@sidra.org (A.A.-K.); kfakhro@sidra.org (K.A.F.); 2Department of Genetic Medicine, Weil Cornell Medical College, Doha P.O. Box 24144, Qatar; 3College of Health and Life Sciences, Hamad Bin Khalifa University, Doha P.O. Box 34110, Qatar

**Keywords:** chromatin, DNA methylation, epigenetics, histone modifications, metaboloepigenetics, miRNA, therapy, type 1 diabetes

## Abstract

Type 1 diabetes (T1D) is an autoimmune condition where the body’s immune cells destroy their insulin-producing pancreatic beta cells leading to dysregulated glycaemia. Individuals with T1D control their blood glucose through exogenous insulin replacement therapy, often using multiple daily injections or pumps. However, failure to accurately mimic intrinsic glucose regulation results in glucose fluctuations and long-term complications impacting key organs such as the heart, kidneys, and/or the eyes. It is well established that genetic and environmental factors contribute to the initiation and progression of T1D, but recent studies show that epigenetic modifications are also important. Here, we discuss key epigenetic modifications associated with T1D pathogenesis and discuss how recent research is finding ways to harness epigenetic mechanisms to prevent, reverse, or manage T1D.

## 1. Introduction

The hallmark of type 1 diabetes (T1D) is T cell-mediated autoimmune destruction of pancreatic beta cells, leading to insulin deficiency, elevated blood glucose concentrations, and life-long need for exogenous insulin therapy. Globally, the annual number of new cases of T1D is rising, and in the under-20 age group alone, numbers are fast approaching 100,000 (www.diabetesatlas.org). The incidence rate varies between countries and ranges from 6% in sub-Saharan Africa to 77% in some parts of Scandinavia [1]; altogether another five million people are expected to be diagnosed with T1D by 2050 [2], with profound implications for healthcare systems globally and the potential for a staggering socio-economic impact. Staying well with T1D requires normoglycemia to be achieved and maintained, but this goal is not currently achievable for many patients using existing treatment strategies [3]. Understanding the intrinsic and extrinsic factors underlying the development and progression of T1D is necessary for the development of novel intervention therapies that could delay or even prevent clinical progression in individuals susceptible to T1D or be used to treat/manage glycemic excursions in those worst affected.

Despite ongoing research, dissecting the etiology of T1D has proven a herculean task. Genetic studies have recognized more than 60 regions (loci) associated with or predisposing to T1D; the first and still strongest reported genetic association is with the human leukocyte antigen HLA region [4,5], while the second most significant T1D genetic association is with insulin gene promoter polymorphism [6]. Even so, only 5% or fewer of children bearing the high-risk HLA haplotypes and about 10% of those with insulin gene polymorphism go on to clinically develop T1D during the first 20 years of their lifetime [7,8]. Therefore, what are the important factors determining whether a genetically susceptible individual develops T1D, and how could these factors be used to help in the fight against this disease?

Epigenetic modification is a key mechanism affecting gene expression, effectively determining the extent and timing of expression of a given gene in response to internal and external stimuli. Multiple lines of evidence now implicate epigenetic factors in T1D incidence: firstly, although monozygotic (MZ) twins share 100% of their DNA, they exhibit a high discordance rate in the development of T1D, especially when disease onset occurs over the age of 15 [9]; furthermore, the rapid increase in T1D seen in recent years is unlikely to be a result of genetic changes across the population alone, due to the short timescale [10,11]. Alongside, other studies showed that in siblings sharing an HLA haplotype, it is rather the age at diabetes onset in their sibling that determines their risk of developing the condition, and not the shared HLA type itself [12]. Thus, it is not only the possession of high-risk genes that decide whether or not an individual progress to the clinical presentation of T1D but also whether, when, and to what extent those genes are expressed.

The growing interest in epigenetic changes in T1D and in factors driving them has led to investigations into epigenetic therapies to treat the condition. In this review, we discuss recent initiatives targeting epigenetic mechanisms involved in T1D, namely, DNA methylation, histone (post-translational) modifications, and non-coding RNA-mediated gene silencing (for a comprehensive discussion of these mechanisms see references [13,14]), as well as highlighting future avenues for productive therapeutic research.

## 2. Therapies Targeting DNA Methylation in T1D

DNA methylation involves the addition of a methyl group to the fifth carbon of CpG dinucleotides (cytosine residues linked by a phosphate to a guanine nucleotide) by DNA methyltransferases, thereby blocking gene transcription [15] (Figure 1). In T1D, there is evidence that DNA methylation in both immune cells and pancreatic islet cells is important and may even substantially pre-date the diagnosis of the disease [16]. Studies of immune cells from the blood of adult MZ twins discordant for T1D have variably identified specific positions with distinct methylation patterns: one study on monocytes found that patterns of methylation at these sites were detectable many years before T1D diagnosis and were correlated with the presence of autoantibodies [17], while a genome-wide DNA methylation analysis identified 88 CpG methylation sites in B lymphocytes with significant modifications, including some affecting genes associated with T1D pathogenesis such as HLA, insulin (INS), interleukin 2 receptor subunit beta (IL-2RB), and CD226 [18]. A more recent MZ twin study did not identify any differences in DNA methylation in monocytes or B cells but found a single differentially methylated CpG position in T cells with genome-wide significance [19].

Genome-wide DNA methylation quantitative trait locus (mQTL) analysis of human pancreatic islets has revealed 383 significant CpG sites in known diabetes loci as potential methylation targets; importantly, some of the identified candidate genes, i.e., glutathione peroxidase 7 (*GPX7*), glutathione S-transferase theta 1 (*GSTT1*), and sorting nexin 19 (*SNX19*) directly affect key biological processes such as proliferation and apoptosis of pancreatic islet beta cells [20]. More recently, Ye et al. identified widespread genetic and epigenetic interactions at known T1D susceptibility loci; they showed (in their preprint) that DNA methylation at five loci, i.e., integrin subunit beta 3 binding protein (ITGB3BP), AF4/FMR2 family member 3 (AFF3), protein tyrosine phosphatase non-receptor type 2 (PTPN2), cathepsin H (CTSH), and cytotoxic T lymphocyte-associated protein 4 (CTLA4), is potentially causal to T1D [21]. Interestingly, it seems that some epigenetic changes may even be present years before T1D diagnosis: Johnson et al. identified 10 regions of the genome in which methylation differs between T1D cases and controls from as early as birth, in some cases multiple years before T1D diagnosis [22]. These findings raise the possibility of screening those with high-risk genetic phenotypes for concurrent high-risk epigenetic marks to identify those at greatest need of intensive monitoring and/or preventative intervention.

Although the significance of altered DNA methylation in T1D onset and pathogenesis is clear, there have yet to be large-scale trials of the use of DNA methyltransferase inhibitors in treating/preventing the condition. However, inhibitors of DNA methylation have been studied in vitro for their potential use in pancreatic cell reprogramming, with the future aim of regenerating patients’ beta cells as a novel T1D therapy. Lefebvre et al. found that treating pancreatic ductal cells with the DNA methylation inhibitor 5-azacytidine successfully promoted their conversion into pancreatic endocrine cells [23]. Similarly, in mice lacking DNA methyltransferase 1(DNMT1) and the alpha cell-maintaining factor Aristaless-related homeobox (Arx) specifically in pancreatic alpha cells, Chakravarthy et al. observed reprogramming of around half of alpha cells into biologically active beta cells in vivo [24]. Interestingly, these authors also found that a subset of T1D patients possess a small number of glucagon-expressing cells lacking both DNMT1 and Arx expression and instead ectopically express beta cell markers, including insulin [24]. While these results suggest that targeted DNMT1 inhibition might support the possibility of regeneration of lost beta cells in individuals with T1D, further studies addressing the longevity and functionality of these converted cells are warranted.

While not specifically assessing epigenetic modification, there are early indications that drugs as well as diets modulating DNA methylation could have some positive effect on delaying the progression to T1D. Long used in diabetes care, metformin has more recently been revealed to have multiple epigenetic modifying actions, including modulating the activity of DNA methyltransferases [25]. The REducing with MetfOrmin Vascular Adverse Lesions (REMOVAL trial) aimed to ameliorate long-term cardiovascular complications in diabetes patients using metformin [26]. In the REMOVAL trial, metformin treatment reduced the progression of atherosclerosis, weight gain, and low-density lipoprotein (LDL)–cholesterol levels [27,28], though the epigenetic effects of the treatment were not specifically measured. In type 2 diabetes, metformin treatment significantly decreased DNA methylation of metformin the transporter genes solute carrier family 22 member 1 (*SLC22A1*), solute carrier family 22 member 3 (*SLC22A3*), and solute carrier family 47 (multidrug and toxin extrusion), member 1 (*MATE1*) in the livers of patients; in the case of *SLC22A1* and *SLC22A3*, reduced methylation was associated with less hyperglycemia and obesity [29]. Similarly, promising results were also seen in pre-clinical trials in adult rats with streptozotocin-induced diabetes treated with the DNMT inhibitor procainamide, which significantly reduced the activity of DNMT in the pancreas and significantly increased fasting insulin levels [30]. Thus, initial studies in both type 1 and 2 diabetes patients and models seem to indicate that DNMT inhibitors might be a promising therapeutic strategy. Several DNMT inhibitors are licensed for use in treating certain types of cancer [31] and could be worth exploring in diabetes.

## 3. Therapies Targeting Histone Modifications in T1D

Alongside direct epigenetic modification of DNA, histones undergo post-transcriptional modifications to restructure chromatin in different ways [32,33,34,35,36]. Genomic DNA is bundled into chromatin, creating nucleosomes; the histone tails project from the nucleosome and are exposed to a wide range of covalent modifications including methylation, acetylation, ubiquitination, phosphorylation, sumoylation, and ADP ribosylation [37,38]. Collectively, these post-transcriptional modifications act to regulate chromatin structure, which has profound effects on several biological activities including gene transcription, DNA repair, and chromosome condensation [32] (Figure 2).

Histone modification is a significant feature of T1D [39], with studies showing aberrant histone modification as well as differential expression of the histone-modifying enzymes histone methyltransferase and histone deacetylase. A pivotal study by Miao et al. revealed increased methylation of lysine 9 in the H3 histone protein (H3K9me2) in lymphocytes from T1D patients: importantly, several of the genes affected had been linked with T1D pathology, such as *CLTA4*, a known T1D susceptibility gene, as well as genes affecting the signaling pathways of the immune mediators transforming growth factor-β (TGF-β), nuclear factor-ĸB (NF- ĸB), p38 mitogen-activated protein kinase (MAPKs), toll-like receptors (TLRs), and interleukin-6 (IL-6) [40]. The same group went on to show that acetylation of lysine 9 in the H3 histone protein (H3K9Ac) upstream of the major T1D susceptibility genes *HLA-DRB1* and *HLA-DQB1* was also significantly higher in monocytes from patients than in those from controls [41]. Furthermore, differential histone methylation and acetylation might be one mechanism explaining the T1D discordance seen in MZ twin studies. Following exposure to high glucose levels, in vitro skin fibroblasts showed significantly different expression of genes regulating epigenetic processes depending on whether they came from the affected or unaffected twin. While the expression of histone lysine methyltransferase (SET 7), H3K4 methyltransferase, and histone deacetylase (HDAC8) was lower in the cells from the type 1 diabetic twin, HDAC 4 was expressed at a higher level [42].

Histone modification inhibitors are a well-characterized class of epi-drugs with clear potential to be used therapeutically for T1D, either alone or in combination with DNA methyltransferase inhibitors. HDAC inhibitors (HDACi) were investigated by Lundh et al. after finding that pancreatic beta cells from children with newly diagnosed T1D exhibited relatively higher levels of HDAC1 and lower levels of HDACs-2 and -3 compared to controls; investigating this further in vitro, the authors showed that inhibition of HDAC1 and -3, but not HDAC2, protected primary and immortalized rat beta cells from apoptosis due to inflammatory cytokine exposure [43]. Building on this work, treating non-obese diabetic mice with the licensed lysine deacetylase inhibitors vorinostat and givinostat reversed diabetes [44], calling for future trials in human patients. Similarly encouraging data have come from trials of valproic acid, an HDAC class I and II inhibitor: in juvenile rats, valproic acid treatment positively influenced beta cell proliferation and functionality and increased insulin production, while simultaneously decreasing beta cell apoptosis and HBA1c and plasma glucose levels [45,46]. Studies in murine models also showed that the small molecule HDAC inhibitor trichostatin A [47] prevented T1D symptoms through increased acetylation of histone H3, which was associated with increased expression of the CD4^+^ T cell-derived lymphokine interferon gamma (INF-γ) and its transcription factor Tbet/Tbx21 [48], highlighting the potential role of chromatin remodeling agents in the protection against the development of T1D.

HDACi have also been used in studies of therapeutic cell reprogramming in the hope that this might be translated into a clinical application. Similarly to DNA methylation inhibitors, in vitro inhibition of the H3K27me3 methyltransferase Ezh2 enhanced neurogenin3-mediated production of beta-like cells from human embryonic stem cells [49], as well as the formation of insulin-producing cells from human induced pluripotent cells [50]. Combining the HDACi romidepsin with the DNA methyltransferase inhibitor 5-azacytidine also led to improved differentiation of adult human dermal fibroblasts into insulin- and glucagon-producing cells [51], but again the therapeutic translation of these findings is so far lacking. Recent studies from Marino and colleagues [52] demonstrate that feeding NOD mice a combination of acetate- and butyrate-rich diets offered protection against T1D progression. These studies indicate that acetate and butyrate might be working through distinct mechanisms and that a beneficial effect may be obtained through a combination of diets rich in these short-chain fatty acids. Acetate decreased the frequency of autoreactive T cells, whilst butyrate increased the number and function of regulatory T cells. Acetate- and butyrate-yielding diets enhanced gut integrity, reduced the circulating concentration of diabetogenic cytokines (such as IL-21), and demonstrated prevention in T1D onset. Clinical trials using such diets will provide intriguing results to understand the timing and potential of such diets in individuals at high risk of T1D.

## 4. Therapies Targeting MicroRNAs in T1D

MicroRNAs (miRNAs) are short (around 22 nucleotides long) non-coding single-stranded RNA molecules [53] that modulate post-transcriptional gene expression, leading to target mRNA silencing (Figure 3). Multiple studies have demonstrated important associations between specific miRNAs and T1D (Table 1). One study on children with recent-onset T1D identified six miRNAs (miR-454-3p, miR-222-3p, miR-144-5p, miR-345-5p, miR-24-3p, and miR-140-5p) with high expression only in the early stages of diabetes; moreover, pathway analysis associated this pattern of differential miRNA expression with glycosaminoglycan biosynthesis as well as with PI3K/Akt, MAPK, and Wnt signaling pathways in early T1D [54]. Similarly, we also demonstrated that enteroviral infections disrupt the miRNA-directed suppression of proinflammatory factors within human islet beta cells, thereby resulting in an exacerbated antiviral immune response that promotes beta cell destruction and eventual T1D [55]. Further research on pancreatic cells from T1D patients linked the expression of miR-23a-3p, miR-23b-3p, and miR-149-5p with that of proapoptotic Bcl-2 proteins and consequent human β cell apoptosis [56], while autoreactive CD8^+^ T cells from patients with T1D showed evidence of repression of pro-apoptotic pathways via increased expression of miR-23b, miR-98, and miR-590-5p [57]. Thus, miRNAs are implicated at every stage of T1D, leading to intense research interest in targeting these molecules therapeutically.

To date, the majority of studies exploring the therapeutic potential of miRNA-targeted treatments for T1D have been carried out in vitro; still, some have yielded promising insights. El Ouaamari et al. showed that the application of 20-*O*-methyl-miR-375 antisense oligonucleotides increased the expression of the target gene PDK1 and reverted insulin release to normal in the rat beta cell line INS-1E [58]. Another study found that blocking miR-21, miR-34a, or miR-146a functions with antisense molecules preserved glucose responsiveness in the murine pancreatic beta cell line MIN6 under treatment with the inflammatory cytokine IL1β and also protected these cells from cytokine-induced apoptosis [59]. Moreover, Lovis et al. showed that blocking miR-34a or miR-146 activity partially protected palmitate-treated MIN6 cells from apoptosis but was insufficient to restore normal insulin secretion [60]. A major challenge of the clinical application of miRNA targeting is delivering the therapeutic agent to the target cells at an effective concentration. Pileggi et al. made important progress in this regard by establishing an islet transplantation model in rats where fluorescently labeled miRNA inhibitors were directly infused via an implanted catheter and effectively localized within target cells [61]. This study paves the way for in vivo trials of miRNA inhibitors aiming to preserve beta cell function or to increase the efficacy of transplantation of autologous islets or of those generated from induced pluripotent stem cells. Building on such work, a recent study conducted by Wang et al. demonstrated the potential of miR-216a to act as a therapeutic target in T1D through its negative regulation of phosphatase and tensin homolog (PTEN) expression levels; the administration of a miR-216a mimic in a murine model of T1D increased beta cell proliferation, decreased PTEN expression, and improved insulin secretion in vivo [62].

Despite these promising results and encouraging progress, many obstacles remain on the path to realizing the potential of miRNA-targeted therapies for T1D. For example, some miRNAs have multiple target genes, and occasionally, multiple miRNAs can regulate the same gene: this complex regulatory network can control several genes via one or a combination of miRNAs, which is further complicated by interactions with other epigenetic factors and linkers, such as DNA methylation and the histone code [63,64]. Future research will require to extend the studies on miRNA, currently carried out at the cellular level, to well-characterized T1D in vivo models to understand the direct role of miRNA-targeted genes and therapies in diabetes treatment.

**Table 1 cells-09-02403-t001:** miRNAs that are differentially expressed between type 1 diabetes (T1D) patients and unaffected controls, targeting the drivers of epigenetic modification in T1D.

miRNA	Target Gene/Pathway Related to T1D	Regulation T1D Patients/Controls	Tissue/Cell	Reference
miR-101-3p	c-MET proto-oncogene, receptor tyrosine kinase (c-Met-HGF), ephrin receptor, and signal transducer and activator of transcription 3 (STAT3) pathways linked to insulin secretion and β cells survival	Upregulated	Serum	[65]
miR-125b-5p and miR-365a-3p	variation of hyperglycemia, measured (HbA1c)	Upregulated	Plasma	[66]
miR-5190 and miR-770-5p	variation of hyperglycemia, measured (HbA1c)	Downregulated	Plasma	[66]
miR-100-5p	Wnt signaling pathway	Downregulated	Serum	[54]
miR-150-5p	transcription factor cMyb and IFN-r	Downregulated	PBMCs	[67]
miR-154-3p	glycosaminoglycan biosynthesis	Downregulated	Serum	[54]
miR-24-3p	MAPK and Wnt signaling pathways	Upregulated	Serum	[54]
miR-25-3p	suppressor of cytokine Ssgnaling 5 (SOCS5), BTG anti-proliferation factor 2 (BTG2), PTEN/Akt pathway, and Notch signaling pathway	Upregulated	Serum	[54]
miR-324-3p	GLI family zinc finger 3 (GLI3), Wnt family member 2B (WNT2B), MAPK pathway, Wnt/β-catenin signaling pathway	Upregulated	Serum	[54]
miR-377-3p	Lysine degradation and ubiquitin-mediated proteolysis pathways	Upregulated	Serum	[54]
miR-378e	Insulin-like growth factor receptor and lipid metabolism pathways, Adiponectin expression pathway	Downregulated	Plasma-derived exosome	[68]
miR-424-5p	caudal type homeobox 2 (CDX2) transcriptional factor pathway	Downregulated	PBMCs	[67]
miR-454-3p	nuclear factor of activated T cells 2 (NFATc2) and vitamin D-dependent calcium-binding protein calbindin 1 (CALB1), Wnt/β-catenin pathways	Upregulated	Serum	[54]
miR-490-5p	Calcium-binding protein 5 (CABP5), phosphatidylinositol-4,5-bisphosphate 3-kinase catalytic subunit alpha (PIK3CA), nuclear factor of activated T cells 5 (NFAT5) and TGF-beta-signaling pathways	Downregulated	Serum	[54]
miR-574-3p	Mothers against decapentaplegic homolog 4(SMAD4) signaling pathway	Downregulated	Plasma-derived exosome	[68]
miR-23a-3p	Death protein 5 (DP5), p53-upregulated modulator of apoptosis (PUMA), BCL2-associated X, apoptosis regulator (BAX), and BIM protein encoded by the *BCL2L11* gene	Downregulated	Pancreatic cells	[56]
miR-23b-3p	DP5 and PUMA	Downregulated	Pancreatic cells	[56]
miR-149-5p	DP5, PUMA, and BAX	Downregulated	Pancreatic cells	[56]
miR-98	TNF-related apoptosis-inducing ligand (TRAIL), TRAIL-receptor 2, FAS cell surface death receptor, and FASLG (Ligand)	Upregulated	T cells	[57]
miR-146a	TNF receptor-associated factor 6 (TRAF6), NUMB endocytic adaptor protein, syntaxin 3 (STX3) and BAF chromatin-remodeling complex subunit(BCL11A)	Downregulated	PBMCs	[67]
miR-590-5p	TRAIL, TRAIL-R2, FAS, and FASLG	Upregulated	T cells	[57]
miR-23b	TRAIL, TRAIL-R2, FAS, and FASLG	Upregulated	T cells	[57]

## 5. Targeting the Drivers of Epigenetic Modification in T1D

While it has long been known that environmental factors can increase the risk of T1D in susceptible individuals, only recently have insights into cellular metabolism revealed epigenetic modifications as one of the mechanistic links between environmental stimuli and autoimmune disease [69]. This insight has created the field of “metaboloepigenetics”, a transformative framework that is connecting the effects of inflammation and nutritional status to metabolite concentration, which in turn influences the enzymes involved in epigenetic regulation, including DNA methyltransferases and HDACs [69]. Studies have now described several mechanisms that are involved in gene expression regulation and genome stability, such as chromatin remodeling, RNA splicing, nuclear RNA export, mRNA stabilization, and non-coding RNA expression, which are strongly influenced by dietary nutrients and related metabolites [70,71,72,73]. Methylation is one of the best described epigenetic modifications whose effects on genome function and relationship to nutrition and other environmental exposures are beginning to be understood [74]. Such knowledge raises the possibility of “metaboloepigenetic reprogramming”: eliminating specific metabolic substrates and managing the accumulation of harmful substances through nutritional and environmental modification is a promising new therapeutic approach to treating complex diseases [75,76,77,78,79]. As such, epigenetic information is a key factor to establish an individual’s inflammation-defense blueprint, whereby an autoimmune disease can be treated through a process of individual categorization, biomarker discovery, and personalized therapeutic intervention [72].

## 6. Conclusions and Future Directions

Diabetes is an expanding global health concern that places a considerable economic burden on individuals and societies and significantly affects patients’ quality of life. Refining the biomarkers of early disease or disease risk and identifying those populations that require intensive monitoring or additional treatment is becoming increasingly important. In conjunction with the advent of precision medicine, studies elucidating the relationship between individual genetic variants and specific epigenetic alterations/epigenetic patterns will provide a pathway for developing effective diagnostic approaches and innovative therapeutic strategies for T1D.

In the last decade, epigenetics has received great attention as the maestro of the T1D orchestra. In this short review, we have described some of the latest discoveries in the field of epigenetics of T1D: how interactions between epigenetic modifications and genetic variants can induce T1D pathogenesis, how altering metabolite concentrations could erase or generate various epigenetic markers and vice versa, how metabolism and chromatin modifications are interconnected, how these processes could significantly influence disease etiology and progression, and how skewing epigenetics can be the key for T1D therapy.

Since T1D is characterized by altered DNA methylation, histone modifications, and expression of several miRNAs, the treatment of this condition based on the restoration of a physiological epigenetic framework is highly conceivable. However, significant challenges remain. Further future study deigns such as nested case–control studies must be done with people with a known risk factor for T1D, as a better understanding of early disease onset will provide insight into the temporal nature of epigenetic alterations and will assess the directivity of their effect, which is currently lacking. Another important aspect is increasing the sample size to provide adequate power, with consideration of ethnicity and population differences. On the other hand, integration of different omics data will reveal new aspects of the disease, for example, allowing investigating the relationship between DNA methylation, alternative splicing, and metabolism in T1D patients. Alongside, some practical barriers need to be overcome, for example, how to deliver epigenetic modifiers to the specific cells and tissues of interest, how to retain them there at therapeutic doses, as well as how to assess the long-term risks, such as off-target effects, of these treatments. Overcoming these challenges will require well-designed and ambitious studies that can link in vitro molecular data with in vivo observations in animal models of T1D and, finally, with clinical studies in diabetes patients.

## Figures and Tables

**Figure 1 cells-09-02403-f001:**
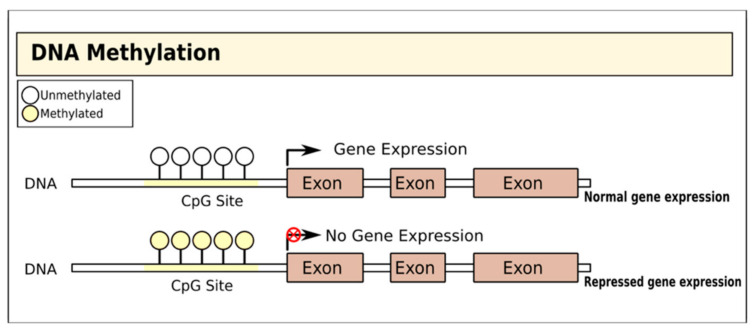
Gene regulation by DNA methylation. DNA is methylated by the covalent addition of a methyl group to CpG dinucleotides by DNA methyltransferases. This process is generally associated with gene silencing.

**Figure 2 cells-09-02403-f002:**
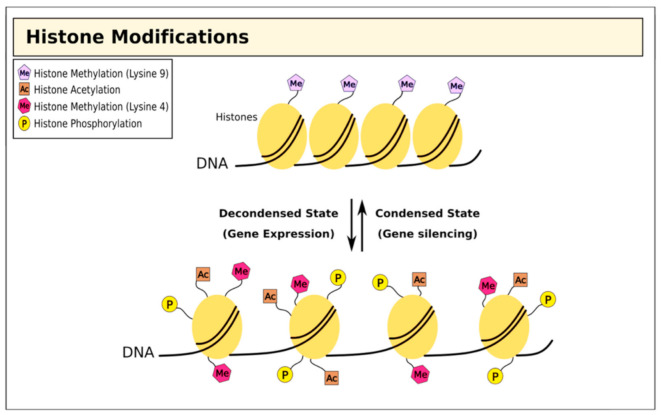
Histone modifications and chromatin structure. Histone methylation at lysine residues is associated with both gene expression and silencing, while acetylation is associated with repression.

**Figure 3 cells-09-02403-f003:**
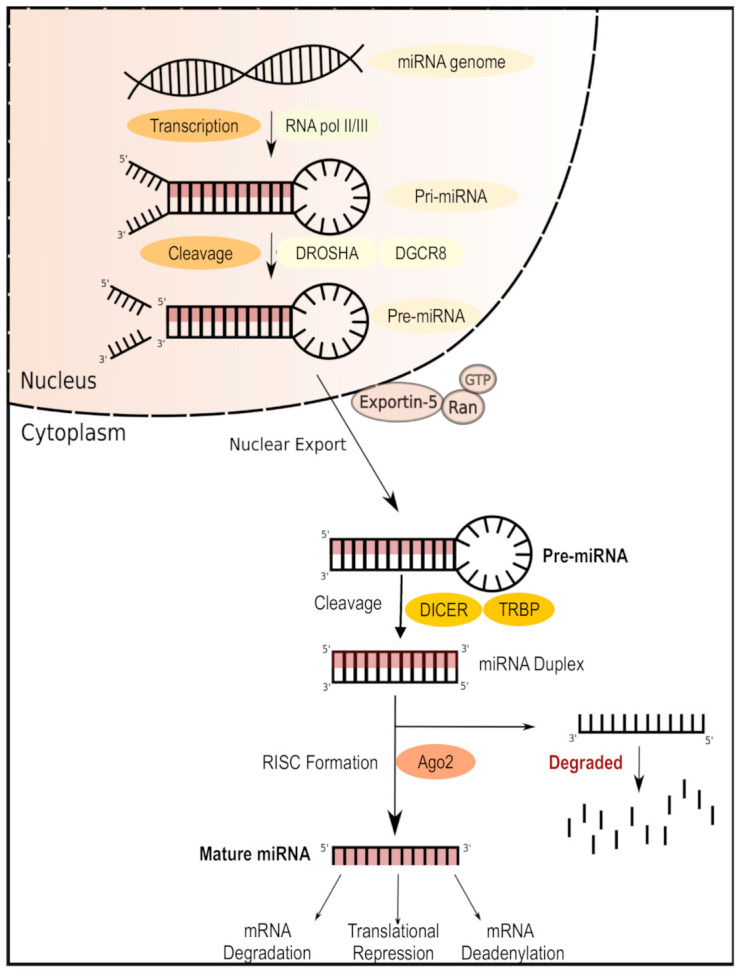
MicroRNA (miRNA) biogenesis and mechanism of action. miRNAs are transcribed in the nucleus as primary miRNA (pri-miRNA) and then cleaved into precursor miRNA (pre-miRNA) by the DORSHA–DGCR8 microprocessor complex. Pre-miRNA is then processed in the cytoplasm by the DICER enzyme, forming a bioactive miRNA duplex which can be bound by the RNA-induced silencing complex ready to target the complementary mRNA for silencing.

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
