# Peer review of "Reading between the (Genetic) Lines: How Epigenetics is Unlocking Novel Therapies for Type 1 Diabetes"

_cells, 2020, doi:10.3390/cells9112403_

Round 1

Reviewer 1 Report

The paper reviews studies that suggest a significant effect of epigenetic modifications on pathogenesis, prevention and treatment of diabetes 1.

The importance of this review is considerable, given the increasing incidence of this pathology and the health and economic effects it entails.

In general, the quality and depth of the study is promising for publication. However, a revision of the following aspects would be advisable:

  • English style and language can be improved with a minor spell check (for example, lines 49, 198, etc.).
  • It would be advisable to provide registered names of genes and loci before their abbreviations for the first time, and not just names or abbreviations independently (for example, lines 82, 86, 106-123, etc.).
  • The text size in Figure 3 should be revised for better viewing.
  • Table 1 is not mentioned in the text.
  • No information is provided on authors' contribution, funding and potential conflict of interest (lines 296-298).
  • The style of bibliographic references should be revised and unified (for example, ref. 4, 7 , etc.

Author Response

Dear Respected Editors/ Reviewers,

The authors are highly appreciating the comprehensive feedback throughout the review process and are
grateful for the endorsement of this manuscript for publication in your journal.

• Comment #1: English style and language can be improved with a minor spell check (for example, lines 49, 198, etc.). English style and language can be improved with a minor spell check (for example, lines 49, 198, etc.).

Authors’ response: The manuscript has now been comprehensively edited for proper English language, grammar, punctuation, spelling, and overall style by qualified native English-speaking editors at Insight Editing London, London, UK.

• Comment #2: It would be advisable to provide registered names of genes and loci before their abbreviations for the first time, and not just names or abbreviations independently (for example, lines 82, 86, 106-123, etc.).

Authors’ response: The manuscript was revised in regards and the full registered names of genes and loci were indicated.

• Comment #3: The text size in Figure 3 should be revised for better viewing.

Authors’ response: Figure 3 was revised as indicated

• Comment #4: Table 1 is not mentioned in the text.

Authors’ response: table has been added to the text as indicated, please see line #196. The table also revised to include all the gene/loci full name ahead of the abbreviation.

• Comment #5: No information is provided on authors' contribution, funding and potential conflict of interest (lines 296-298).

Authors’ response: The requested information was added, please see line 298-303.

• Comment #6: The style of bibliographic references should be revised and unified (for example, ref. 4, 7 , etc.

Authors’ response: The manuscript was revised and adjusted according to this comment.
o All revised text is highlighted in RED

Reviewer 2 Report

This is an excellent review discussing epigenetic modifications associated with type 1 diabetes pathogenesis.  Recent research findings could be of assistance to develop strategies to use epigenetic mechanisms in the prevention and treatment of type 1 diabetes. the paper will be of great interest to physicians and scientists in the field. 

Author Response

Dear Respected Editors/ Reviewers,

The authors are highly appreciating the comprehensive feedback throughout the review process and are grateful for the endorsement of this manuscript for publication in your journal.